# Imaging Modalities Used for Frameless and Fiducial-Less Deep Brain Stimulation: A Single Centre Exploratory Study among Parkinson’s Disease Cases

**DOI:** 10.3390/diagnostics12123132

**Published:** 2022-12-12

**Authors:** David Krahulik, Filip Blazek, Martin Nevrly, Pavel Otruba, Lumir Hrabalek, Petr Kanovsky, Jan Valosek

**Affiliations:** 1Department of Neurosurgery, University Hospital Olomouc, 77900 Olomouc, Czech Republic; 2Department of Neurology, University Hospital Olomouc, 77900 Olomouc, Czech Republic

**Keywords:** Parkinson’s disease, deep brain stimulation, Nexframe, O-arm

## Abstract

Deep brain stimulation (DBS) is a beneficial procedure for treating idiopathic Parkinson’s disease (PD), essential tremor, and dystonia. The authors describe their set of imaging modalities used for a frameless and fiducial-less method of DBS. CT and MRI scans are obtained preoperatively, and STN parcellation is done based on diffusion tractography. During the surgery, an intraoperative cone-beam computed tomography scan is obtained and merged with the preoperatively-acquired images to place electrodes using a frameless and fiducial-less system. Accuracy is evaluated prospectively. The described sequence of imaging methods shows excellent accuracy compared to the frame-based techniques.

## 1. Introduction

Deep brain stimulation (DBS) is a widespread technique used to neuromodulate subcortical brain structures in patients with neurological and psychiatric disorders, mainly Parkinson’s disease (PD), dystonia, essential tremor, obsessive-compulsive disorder, and schizophrenia. The highest level of evidence supports using this method in PD compared to the best pharmacological treatment [1]. DBS is now indicated sooner and more frequently, even for the earlier stages of PD. The neurosurgical techniques used to place the electrodes range from frame-based stereotaxis with microelectrode recording and physiological mapping of target structures [2,3] to a frameless implantation technique using neuro-navigation guidance with skull-mounted aiming devices, in conjunction with bone-implanted fiducial markers.

The success of the surgical treatment is primarily dependent upon the proper placement of the electrodes in the brain. Anatomical structures used as targets, such as the thalamus or subthalamic nucleus (STN), could be localised based on structural images from magnetic resonance imaging (MRI). However, only whole structures are depicted in these structural images. Therefore, it is not possible to distinguish their parts with these methods (e.g., parts of the STN) or nuclei (e.g., ventral intermedial nucleus of the thalamus), which are crucial for precise targeting in patients with essential tremor and tremor in PD. Diffusion-weighted imaging (DWI/dMRI) can disclose anatomical connectivity using tractography, visualizing thalamo-cortical pathways [4,5] and providing dMRI-based thalamic parcellation. Some have established probabilistic tractography-based parcellation as an acceptable and reliable method to separate the ViM nucleus. In STN, the subdivisions can also be identified using the connectivity-based parcellation by the hyper-direct pathway projections [6]. The motor (lateral) part of the STN is the desired target for DBS in PD; whereas, stimulation of the limbic part is responsible for some of the adverse, non-motor effects [7,8].

The current clinical practice uses a structural (T2-weighted) MRI to define the target, which is being done by an experienced clinician. The repeated proposal to use tractography-based parcellation to improve STN targeting has provided inconsistent results, and the clinical use of this method remains a matter of ongoing debate. Therefore, the use of MRI tractography is our primary method used to visualise the white matter tracts in vivo, representing the anatomical connectivity of the STN, identifying the parts responsible for motor and non-motor symptoms. To perform a frameless and fiducial-less DBS procedure, it is necessary to use other imaging methods and fuse them using a planning station. In our centre, we use a CT scan, which is fused to an MRI and for intraoperative imaging and an O-arm scan, which is fused to the CT scan. Our project aims to present the numerous imaging techniques required for the highest possible efficiency and efficacy of placing DBS electrodes into the STN in PD patients, using tractography as guidance. The final placement of the DBS electrodes is the main factor influencing a successful outcome of DBS, conjoining with an appropriate selection of candidates, optimised programming, and pharmacotherapy after the surgery.

## 2. Materials and Methods

The use of a combination of imaging modalities is needed for the best possible outcome of surgical placement of DBS electrodes. Thus, the precise knowledge of the STN region is crucial in achieving the most significant antiparkinsonian benefit. This precise localisation can only be achieved by the accurate and effective use of ideal imaging techniques. 

The best clinical effect on motor symptoms is based on the optimal localisation of the lateral part of the STN, which also leads to the smallest possible number of side effects of the surgical therapy induced by stimulation of adjacent neuroanatomical structures. For evaluation of the clinical impact of DBS therapy in PD patients, the Unified Parkinson’s Disease Rating Scale (UPDRS) parts III and IV are used, in conjunction with the levodopa equivalent daily dose (LEDD) [9,10]. All the above evaluations are made preoperatively in OFF medication and half a year after the surgery in OFF medication with ON stimulation. OFF-medication is defined as refraining from antiparkinsonian pharmacotherapy for at least 12 h or one day for long-release formulations. By retrospective assessment of the structural connectivity based on the position of the optimal active contacts, the volume of tissue activated (VTA) enables us to further refine the method by identifying markers predicting good treatment response and side effects of the therapy.

Acquiring all this information should also lead to an improved understanding of the pathways that connect subcortical and deep structures of the brain.

Patients were included based on meeting the two main criteria–confirmed Parkinson’s disease and suitability for surgery. Patients that did not meet the MDS-PD criteria were excluded from this study. For patients that met the criteria, comorbidity, compliance, and life expectancy were evaluated. Patients who were not compliant enough to cooperate for the 3-h implantation done under local anaesthesia were excluded, as were those unable to go under general anaesthesia for the second phase of the surgery, implantation of the Implantable Pulse Generator. All other patients passing these criteria were approved for surgery and included in our study.

In total, 11 patients were included in our study. The average age was 64 (57 to 75) years old. There were 6 females and 5 males; all patients were Caucasian. 

### 2.1. Imaging Methods

#### 2.1.1. MRI Examination

MRI examination before the surgery is performed on 3T MR scanner (Siemens Vida, Erlangen, Germany) using a 20-channel head/neck coil seven days before the surgery. The protocol contains structural sequences and dMRI sequences used for the tractography.

##### Diffusion Magnetic Resonance Imaging (dMRI) Procedure 

Diffusion MRI (dMRI) is a technique sensitive to random water molecule movement in tissues. Various models can mathematically model the directionality of the diffusion. This information can then be used for a method called tractography, which identifies white matter tracts and bundles in-vivo. Advanced diffusion models such as Ball-and-Sticks [11] allow modelling diffusion within multiple compartments to characterise diffusion properly in voxels with more than one fibre population, which is crucial for identifying subsidiary pathways [12]. Multi-compartment models require high-angular resolution diffusion imaging (HARDI). The 3T HARDI protocol, in combination with similar techniques like simultaneous multi-slice, can acquire high-resolution dMRI data covering >64 diffusion directions within a clinically acceptable time of about 5 min. There are no requirements for examined patients to cooperate during dMRI examination besides keeping their head still. Cushions immobilise the subject’s head to assure maximum comfort and minimise head motion.

##### Data Acquisition

The MR imaging protocol covers the whole brain with focusing on deep brain structures, such as the thalamus and STN, and consists of a high-resolution T1-w 3D MPRAGE (Magnetization Prepared-RApid Gradient Echo) of the whole head (0.9 mm^3^ isotropic voxel, 256 × 256 matrix, 208 slices) with and without a contrast agent (Gd-based), T2-w 2D axial TSE (Turbo Spin Echo) sequence (0.4 × 0.4 × 2.4 mm^3^, 448 × 448 matrix, 30 slices) scan, and T2-w 2D axial SWI (Susceptibility Weighted Imaging) sequence (0.9 × 0.9 × 1.5 mm^3^, 232 × 256 matrix, 80 slices). (Figure 1). Diffusion data are collected utilising the HARDI technique to equally sample diffusion gradients with reversed phase-coding (anterior-posterior (A-P) and posterior-anterior (P-A)), resulting in pairs of images with distortions going in opposite directions (2.0 mm^3^ isotropic voxel, 100 × 100 matrix, 60 slices, 64 diffusion directions b = 1000 s/mm^2^ with interspersed b = 0 s/mm^2^ image every 8 diffusion volumes in A-P direction and 8 b = 0 s/mm^2^ images in P-A direction). The duration of the entire MRI examination is about 30 min.

##### Analysis of MRI Data, Tractography, and STN Segmentation

The patients in our study were divided into two groups; in the active group (6 patients), the neurosurgeon adjusted the preoperative virtual plans of electrode placement according to the tractography-based STN segmentation. In the control group (5 patients), tractography is performed retrospectively, but no correction of electrode placement is done, and the surgery team remains blinded to the STN parcellation.

Diffusion MRI data processing is done using FSL (FMRIB Software Library, www.fmrib.ox.ac.uk/fsl, accessed on 1 January 2020); structural MRI data processing is performed using FSL and FreeSurfer (http://surfer.nmr.mgh.harvard.edu/, accessed on 1 January 2020). Pre-processing of dMRI data consists of correcting motion artefacts, susceptibility artefacts, and eddy currents artefacts using FSL’s *topup* and *eddy* tools based on reversed phase-coding sequences. Multi-compartment Ball-and-Stick model is then fitted on artefacts-corrected dMRI data using the *4edpost* [11] function. Initially, STN is segmented using a standard anatomical atlas [13] complemented with manual edits [14,15,16] if necessary, exploiting T_2_-weighted and SWI images linearly co-registered with high-resolution T_1_-weighted image using the FLIRT tool [17]. Next, the neurosurgeon creates virtual manual plans of electrode placement for both patient groups (while blinded to group allocation). Brodmann areas (BA) 4, 6, and 1–3, corresponding to the primary motor cortex, premotor cortex, and primary somatosensory cortex, respectively, are extracted from the cortical reconstruction. Volumetric segmentation is performed using FreeSurfer and used as seed areas for the hyper-direct pathway to the motor part of the STN. In contrast, BA 25, 30, 34, and 35 serve as seed areas for the pathway reaching the limbic region of the STN [18]. Using the probtrackx [11,16,18] tool, probabilistic tractography is performed from cortical seed areas to STN to delineate the two STN subdivisions. Time-demanding estimation of the Ball-and-Sticks model and running of probabilistic tractography is parallelised using the HTCondor parallelisation tool over multiple CPUs. The resulting parcellation of STN is merged with the MPRAGE image, imported to the planning station (Medtronic, Minneapolis, MN, USA), and co-reregistered using rigid registration (6 degrees of freedoms–3 translations, 3 rotations) with the reference CT.

#### 2.1.2. CT

A pre-surgical CT scan covering the whole head is acquired on a clinical machine (GE LightSpeed VCT) using the helical mode several days before the surgery. Images from the CT are then imported into the planning station (Medtronic, Minneapolis, MN, USA) and used as reference images for consequent MRI and O-arm data registrations. Acquisition of this image is crucial for perfect alignment with the MRI and intraoperative CT.

#### 2.1.3. Intraoperative CT

Intraoperative 3D cone-bean CT (Medtronic O-arm O2, Minneapolis, MN, USA) covering the whole head is acquired at the beginning of the surgery using the Stereotaxic mode and 40 cm FOV (field-of-view). The gantry of the O-arm is positioned to have the patient’s head in the isocentre. If necessary, the gantry can be tilted to achieve the best possible alignment with the patient’s head. After the scan is finished, the exact scanning position of the gantry is saved, and the gantry is moved horizontally towards the patient’s legs to free up space for the surgeon. The acquired 3D image is automatically transferred to the neuro-navigation Stealth Station S8 (Medtronic, Minneapolis, MN, USA) and co-registered using rigid registration (6 degrees of freedoms–3 translations, 3 rotations) with the reference preoperative CT image to allow precise intraoperative navigation.

Three O-arm scans are made during the surgery in total. The first O-arm scan is performed at the beginning of the surgery to localise entries for the electrodes. The second O-arm scan is performed after attaching the reference frame (NexFrame, Medtronic, Minneapolis, MN, USA) to the right side of the patient’s head. The third O-arm scan is performed after attaching the reference frame to the left side of the patient’s head.

### 2.2. Surgical Technique

The surgery is performed in one day, split into two stages. During the first stage—insertion of the DBS electrodes—the patient is awake; the second stage—implantation of the internal pulse generator—is done under general anaesthesia.

An intraoperative CT scan using 3D O-arm is acquired at the start of the surgery, fused with MRI and CT images made preoperatively, using the Stealth Station S8 stereotactic navigation software. The selection of target points for the tips of the electrodes is made using a combination of direct and indirect targeting in PD. Using the volumetric MRI images, the trajectories are shown, and minor adjustments are made to the trajectories to avoid damaging dural venous sinuses, cortical veins, and lateral ventricles. (Figure 2)

The burr hole entry point of the predetermined electrode trajectory is localised and marked on the skin using a passive planar blunt probe and active S8 navigation; a small hole is drilled to mark that point on the skull. After that, sterile draping is done, a skin incision is made, and the burr holes centred on the pilot holes are completed. With the lead anchoring device and the Nexframe base attached, a navigated O-arm picture is acquired and fused with preoperative imagery. Registration of sterile instruments is performed to achieve a registration error under 0.5 mm. Attachment and alignment of the Nexframe tower are done using S8 navigation software to correspond to the chosen target. The target depth is then calculated and set on the microTargeting Drive System positioning device. The dura mater is closed to prevent CSF leak or pneumocephalus with the help of fibrin glue.

### 2.3. Intraoperative Microelectrode Registration (MER)

Four needles are used to perform MER in STN-DBS, allowing us to delineate the borders of the STN using an array as central, lateral, anterior, and posterior. The starting point for STN is set 10 mm above the MRI based localisation, and the Microdrive makes an advancement of 500 μm towards the target each time.

### 2.4. Macro-Test Stimulation

After the microelectrode registration is done, the microelectrode tip is retracted. Channels that showed clinically significant activity over a length exceeding 3 mm are selected for intraoperative, real-time test stimulation (for PD, the duration of the pulse is 60 ms, the frequency of the pulse is 130 Hz, 1–4 mA) [19]. The chosen electrode with the macro-tip is then used for macro-test stimulation, performed by an experienced neurologist. After evaluating the selected channels, the one with the largest therapeutic window, meaning the lowest current threshold for the highest clinical response and the highest threshold for side effects, is chosen for permanent electrode implantation. Afterwards, the final control 3D O-arm scan is performed to confirm the accuracy of electrode placement.

### 2.5. Clinical Assessment

Both motor and non-motor symptoms (NMS) were evaluated preoperatively and postoperatively at months 1 and 4, utilising a large variety of scales.

For motor symptoms, we used the Movement Disorder Society Unified Parkinson’s Disease Rating Scale, (MDS-UPDRS), part III: Motor Examination.

The main scale used for non-motor symptoms was the Nonmotor Symptoms Scale for Parkinson’s Disease (NMSS), which tests the severity and frequency of NMS over the previous 30 days. Other scales used were The Parkinson’s Disease Sleep Scale (PDSS), The Parkinson’s Disease Questionnaire (PDQ-39), and the *Scales for Outcomes* in Parkinson’s disease—Autonomic (SCOPA-Aut) Questionnaire.

## 3. Results

For all 11 patients, we implanted four microelectrodes using the parallel multi-track microelectrode recording. Afterwards, we confirmed the optimal position of the electrodes within the STN by recording the signal from the nucleus in all patients. On average, three microelectrodes had a good signal (one minimum, four maximum). After MER is finished, the tip of the microelectrode was retracted. Channels showing clinically significant activity over a length exceeding 3 mm are selected for intraoperative, real-time test stimulation (for PD, the pulse duration was 60 microseconds, the frequency of the pulse was 130 Hz, 1–4 mA).

To confirm the proper position of the lead and the accuracy of the procedure, we obtained an intraoperative 3D O-arm scan and fused the newly acquired image with preoperative planning. This can be used anytime during the surgery to control the procedure (Figure 3).

The precision of insertion of the electrode into the STN can be evaluated by calculating error on the pre-/perioperative MRI/CT fusion images. The point of entry anterior commissure-posterior commissure (AC-PC) coordinates point A and the target point AC-PC coordinates point B of the trajectory can be found on the perioperative navigation device using preoperative MRI. The target is then usually intraoperatively modified in accordance with the micro recording and clinical examination by shifting the trajectory, labelled as distance d. Knowing both the distance and the AC-PC coordinates of both the starting point and the planned target enables us to calculate the AC-PC coordinates of the modified target (point C). The electrodes’ actual position AC-PC coordinates (point D) can be localised by manually placing a cursor at the end of the electrode, which is visible on the MRI/CT fusion using the navigation device. Using the equation to calculate the distance between two points in three-dimensional space makes it possible to determine the total error (distance between the actual placement position and the modified target). Using an equation for calculating the distance of two points in a straight line, we can quantify the placement error in anteroposterior, lateral, and vertical axes. The accuracy of placement is shown in Table 1. We found no significant differences by comparing our results to other publications [1,10].

The accuracy of electrode placement adjusted in accordance with pre-operative STN parcellation (active group) has been slightly better in comparison to the control group; however, these results were not statistically significant.

### 3.1. Surgical Outcomes and Complications

Patients treated with DBS for PD have reported significant improvements in motor fluctuation, documented in the patient’s diaries before the surgery and six months after DBS STN. Patients mark into their diaries one of three states (OFF state, ON without dyskinesias, or ON with dyskinesias) every hour for three consecutive days before the DBS procedure and then for three successive days six months after the DBS. The mean OFF time (in hours) before the DBS was compared to the mean OFF time after the DBS. Six months after DBS, the mean OFF time was reduced by 52%; dopaminergic medication was reduced by 54.3%.

One patient undergoing frameless, fiducial-less STN-DBS had an infection post-operatively that ultimately led to the removal of the entire system. The patient then underwent reimplantation with no further complications. Postoperative CT scans showed bilateral subdural air with no clinical symptoms in two patients. No other complications were noted in the clinical records of the rest of the patients.

### 3.2. Clinical Outcomes

The motor symptoms have improved significantly at months 1 and 4, as expected, as STN-DBS has been primarily developed for the treatment of PD motor complications.

For non-motor symptoms, the overall NMSS values have improved at both month 1 and 4, with mainly the gastrointestinal, urinary, and cardiovascular sub-scores reduced significantly.

## 4. Discussion

Two main methods are nowadays used to perform DBS, one using a stereotactic frame and the other using a frameless system with small fiducials attached to the skull [20]. (Figure 4). Excluding the fiducials in this new method and utilising perioperative O-arm imaging and an online navigation system does not decrease the accuracy. None of the existing techniques achieve perfect accuracy of electrode placement, and the average error is between 1 and 2 mm. Several weak points can lead to inaccuracy with this new method, mainly CT, MRI, and O-arm fusion. Still, by using the latest navigation system, the error lowers to only about 1 to 2 imaging voxels.

Urgosik et al. [21] analysed the weak points that can lead to inaccuracy with this method, mainly the fusion of MRI, CT, and O-arm, the accuracy of DBS placement using the Leksell frame according to intraoperative monitoring. Still, the newest navigation system has an error of about 1–2 imaging voxels. Rohlfing et al., on the other hand, analysed the accuracy of DBS placement using the Leksell frame according to intraoperative CT, which had a reduced accuracy of stereotactic frames because of torque introduced by the effect of weight-bearing monitoring with excellent results and minimum complication. Holloway et al. [22] and Krahulik et al. [23] confirmed the comparable accuracy of the frameless approach, which pointed to reduced accuracy of stereotactic frames because of torque introduced by the effect systems to the frame-based systems.

Articles published recently focusing on the O-arm stereotactic registration precision show good procedure accuracy if operated by experienced Nexframe users. Our study has shown similar results, but the number of patients is smaller. Using tractography based STN parcellation preoperatively shows higher accuracy, but the results are limited by a relatively small group of patients in our study and more work is needed to prove this point.

The focus of this study was not surgery times or patients’ toleration of the overall procedure. Still, having used all three methods in our centre, we can conclude that the patients best tolerate this new procedure, and it shortens the operation time. Disadvantages of this procedure could be the radiation dose during multiple perioperative O-arm scanning for the patient and a longer learning curve to achieve accuracy for the surgeon and his team. All our patients indicated for DBS as a treatment for PD, dystonia, or ET are currently operated on with this procedure [24]. For some non-PD diagnoses, such as those mentioned above, the spectrum of imaging techniques can be narrower as the anatomical targets can be seen on native MRI scans in a quality sufficient for exact electrode placement.

## 5. Conclusions

Our single centre exploratory study has proved that the frameless, fiducial-less method using the perioperative O-arm and Nexframe navigation system is an equally accurate and safe procedure that has been well-tolerated by all our patients. The accuracy of electrode placement is comparable to recently published articles, and our previous work focused on a frameless Nexframe approach. As we obtain a larger group of patients in the future, we can convincingly prove our initial findings to be correct. This method should be primarily used by a team with experience with the Nexframe system.

## Figures and Tables

**Figure 1 diagnostics-12-03132-f001:**
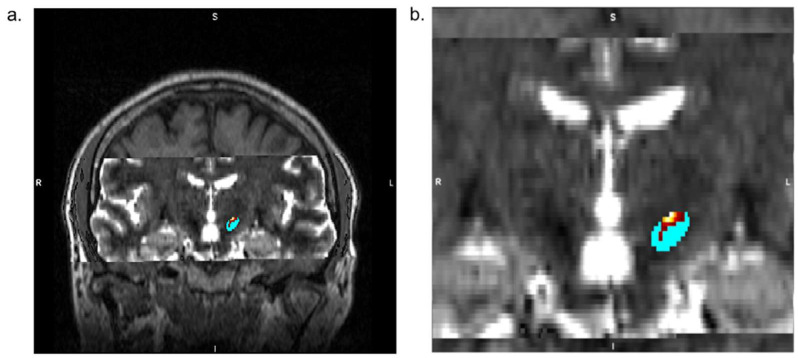
Example of STN parcellation: (**a**) Coronal view of T1-w MPRAGE overlaid by T2-w image with adjusted brightness and contrast to delineate STN. The light-blue colour indicates a segmented mask of STN, and the yellow-red colour indicates the motor part of STN identified by probabilistic tractography. (**b**) Zoomed detail to STN in relation to lateral ventricles.

**Figure 2 diagnostics-12-03132-f002:**
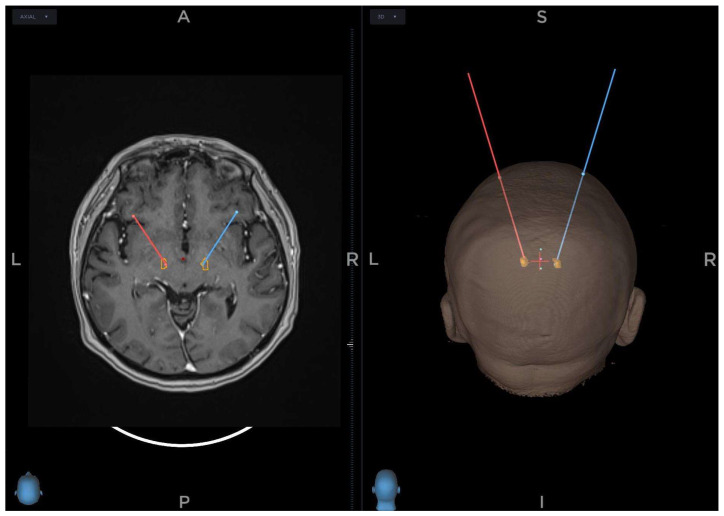
Planning trajectories and fusion of the images on the planning station (red line—trajectory on the left side, blue line—trajectory on the right side, A—anterior, P—posterior, R—right, L—left, S—superior, I—inferior, yellow—subthalamic nucleus).

**Figure 3 diagnostics-12-03132-f003:**
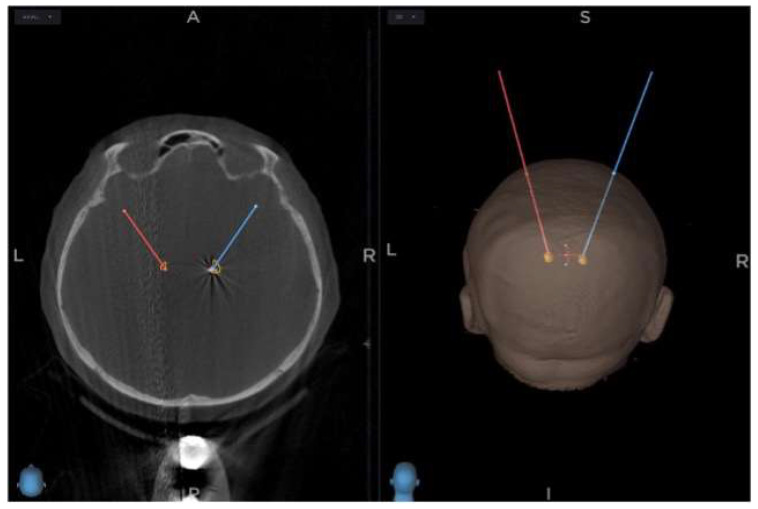
Final position of the lead within the STN nucleus on the right side controlled intraoperatively with O-arm scan fused to preoperative plan (red line—trajectory on the left side, blue line—trajectory on the right side, A—anterior, P—posterior, R—right, L—left, S—superior, I—inferior, yellow—subthalamic nucleus).

**Figure 4 diagnostics-12-03132-f004:**
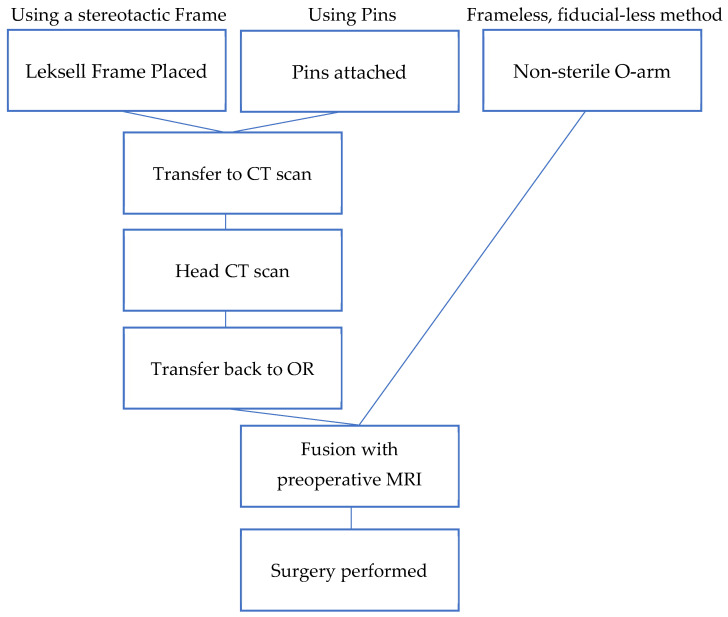
Flowchart comparison of older stereotactic methods with frameless, fiducial-less method.

**Table 1 diagnostics-12-03132-t001:** Accuracy of the Frameless Fiducial-less procedure.

Procedure	Total Error (mm)	Lateral Axis	AP Axis	Vertical Axis
FL	1.79 ± 0.68	1.10 ± 0.78	1.37 ± 0.87	1.21 ± 0.9

## Data Availability

The datasets used and/or analysed during the current study are available from the corresponding author on reasonable request.

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
