# Peer review of "Imaging Modalities Used for Frameless and Fiducial-Less Deep Brain Stimulation: A Single Centre Exploratory Study among Parkinson’s Disease Cases"

_diagnostics, 2022, doi:10.3390/diagnostics12123132_

Round 1

Reviewer 1 Report

This is an interesting and well-presented study of DBS. The authors described their set of imaging modalities used for a frameless and fiducial-less method of DBS. STN parcellation was done based on diffusion tractography. Electrodes were placed using the intraoperative O-arm-controlled frameless and fiducial-less system. This method has been proved to be equally accurate, safe and well-tolerated.

The weakness of this study was the relatively small sample size, and no comparison with other procedures with stereotactic frame or using bone-implanted fiducial markers.

1. The general information of the patients (or the patient demographic data) should be included in the part of "Materials and Methods".

2. The patients in the study were divided into two groups: the active group, and the control group (line 132).

How many patients in each group were included in the analysis, and the results from these two groups should be explained and analyzed.

3. The syntax of the literature references is not uniform (eg. Ref 17, 23, 26, etc.). Please follow the appropriate model for the reference and more proofreading is needed.

Author Response

Response to Reviewer 1 Comments

This is an interesting and well-presented study of DBS. The authors described their set of imaging modalities used for a frameless and fiducial-less method of DBS. STN parcellation was done based on diffusion tractography. Electrodes were placed using the intraoperative O-arm-controlled frameless and fiducial-less system. This method has been proved to be equally accurate, safe and well-tolerated.

The weakness of this study was the relatively small sample size, and no comparison with other procedures with stereotactic frame or using bone-implanted fiducial markers.

Point 1: The general information of the patients (or the patient demographic data) should be included in the part of "Materials and Methods".

Response 1: Thank you for this point, we have added basic general information about our patients into the Material section.

Point 2: The patients in the study were divided into two groups: the active group, and the control group (line 132).

How many patients in each group were included in the analysis, and the results from these two groups should be explained and analyzed.

Response 2: We thank you for this comment, we have added the missing information, and have further clarified the results in the Results and Discussion sections.

Point 3: The syntax of the literature references is not uniform (eg. Ref 17, 23, 26, etc.). Please follow the appropriate model for the reference and more proofreading is needed.

Response 3: Thank you, we have parsed and corrected the list of references.

Reviewer 2 Report

In this study, the authors applied a set of imaging modalities to define the optimal localization of STN for DBS within 11 PD patients. They demonstrated this frameless, fiducial-less method is an equally accurate and safe procedure. While, the following part should be revised:

1.    It is important to describe how patients are included and excluded from the study.

2.    Pre- and post-outcomes of patients should be detailed, and both motor and non-motor symptoms should be compared. Moreover, it is unclear whether the active group's outcome differed from the control group's.

3.    Applied sequence imaging acquisition should be more detailed, especially the voxel size, which affects image resolution.

4.    As only T1wi and T2wi images were shown in figure1, illustrating STN parcellation, it was unclear whether all mentioned sequences were used for targeting? What role do other sequences play in STN parcellation?

5.    In what sense does setting the pathway reaching the limbic region of the STN as contrast serve? In this area, do authors also place microelectrodes?

It made me feel confused about the conclusion since the methods and results part was not organized well.

Author Response

Response to Reviewer 2 Comments

In this study, the authors applied a set of imaging modalities to define the optimal localization of STN for DBS within 11 PD patients. They demonstrated this frameless, fiducial-less method is an equally accurate and safe procedure. While, the following part should be revised:

Point 1: It is important to describe how patients are included and excluded from the study.

Response 1: Thank you for this suggestion. We have added the inclusion and exclusion criteria to the Material and Methods section of the article.

Point 2: Pre- and post-outcomes of patients should be detailed, and both motor and non-motor symptoms should be compared. Moreover, it is unclear whether the active group's outcome differed from the control group's.

Response 2: We thank you for this suggestion and have added basic data and results for both motor and non-motor symptoms. However, the main premise of our study was to show which imaging modalities we use for Frameless and Fiducial-Less Deep Brain Stimulation and to prove and compare the accuracy and safeness of this method. Going into detail about patient clinical outcomes would divert the study from our original plan.

Point 3: Applied sequence imaging acquisition should be more detailed, especially the voxel size, which affects image resolution.

Response 3: We thank the reviewer for this suggestion. We detailed the sequence parameters in the Methods section.

Point 4:  As only T1wi and T2wi images were shown in figure1, illustrating STN parcellation, it was unclear whether all mentioned sequences were used for targeting? What role do other sequences play in STN parcellation?

Response 4: Thank you for this relevant point. As described in the Methods section in the chapter Analysis of MRI data, tractography and STN segmentation, SWI and T2-w images are used for subthalamic nucleus identification. Diffusion MRI scans are utilized for tractography-based subthalamic nucleus parcellation. And T1w 3D MPRAGE image is the high-resolution reference scan where tractography results are registered.

Point 5: In what sense does setting the pathway reaching the limbic region of the STN as contrast serve? In this area, do authors also place microelectrodes? It made me feel confused about the conclusion since the methods and results part was not organized well.

Reponse 5: We thank the reviewer for this question. We place microelectrodes only into the motor part of the STN. The limbic part of the STN is identified to assess its location relative to the motor part.

Round 2

Reviewer 2 Report

My questions had been answered and the manuscript had been improved by the authors. In addition, I recommend adding a flowchart to clarify the steps of the new approach described in the study, as well as a comparison with the original approach. It may be helpful for readers to understand the study's aim and methods.

Author Response

Thank you for this comment about our study. We agree that adding a flowchart will help clarify the benefits of using our new method in comparison to the original approaches.